# Structural Perspectives on the Mechanism of Soluble Guanylate Cyclase Activation

**DOI:** 10.3390/ijms22115439

**Published:** 2021-05-21

**Authors:** Elizabeth C. Wittenborn, Michael A. Marletta

**Affiliations:** California Institute for Quantitative Biosciences, Departments of Chemistry and of Molecular and Cell Biology, University of California, Berkeley, CA 94720, USA; ecwittenborn@berkeley.edu

**Keywords:** nitric oxide, soluble guanylate cyclase, cryo–electron microscopy, enzyme structure

## Abstract

The enzyme soluble guanylate cyclase (sGC) is the prototypical nitric oxide (NO) receptor in humans and other higher eukaryotes and is responsible for transducing the initial NO signal to the secondary messenger cyclic guanosine monophosphate (cGMP). Generation of cGMP in turn leads to diverse physiological effects in the cardiopulmonary, vascular, and neurological systems. Given these important downstream effects, sGC has been biochemically characterized in great detail in the four decades since its discovery. Structures of full-length sGC, however, have proven elusive until very recently. In 2019, advances in single particle cryo–electron microscopy (cryo-EM) enabled visualization of full-length sGC for the first time. This review will summarize insights revealed by the structures of sGC in the unactivated and activated states and discuss their implications in the mechanism of sGC activation.

## 1. Introduction

Soluble guanylate cyclase (sGC) is a nitric oxide (NO)-responsive enzyme that serves as a source of the secondary messenger cyclic guanosine monophosphate (cGMP) in humans and other higher eukaryotes [1]. Upon NO binding to sGC, the rate of cGMP formation increases by several hundred-fold, effectively amplifying the initial NO signal. cGMP in turn regulates downstream effectors, such as cGMP-dependent protein kinases (PKGs), ion channels, and phosphodiesterases (PDEs), leading to a variety of physiological responses that include vasodilation, inhibition of platelet aggregation, and effects on neurotransmission [2,3]. As a result, dysregulation of the NO-sGC-cGMP signaling pathway is associated with various disease pathologies, including hypertension, cardiovascular diseases, neurodegenerative diseases, and asthma [4,5,6,7,8]. Indeed, stimulators of sGC activity have proven effective in the treatment of pulmonary arterial hypertension and chronic thromboembolic pulmonary hypertension. The first sGC-targeted drug, Adempas^®^, was approved by the US Food and Drug Administration in 2013 [9]. Given this clinical relevance, sGC has received considerable attention over the years, with research efforts aimed at better understanding the complex mechanism of activity regulation. Biochemical aspects of sGC activation and deactivation have been reviewed recently [10] and this review will focus on recent and exciting advances in our knowledge of sGC from a structural perspective.

## 2. Domain Architecture of sGC

sGC is an obligate heterodimer consisting of homologous α and β subunits that share roughly 30% sequence identity, depending on the species of origin. Each subunit contains four domains: an N-terminal heme NO/O_2_-binding (H-NOX) domain, followed by a Per-Arnt-Sim (PAS) domain, a coiled coil (CC) domain, and a C-terminal catalytic (CAT) domain (Figure 1A). As its name implies, the H-NOX domain contains a heme cofactor that binds a single equivalent of NO at the iron center [11,12,13]. Although both α and β subunits contain an H-NOX domain, only the β H-NOX domain contains a heme [12,14]. PAS domains are widely found in signaling proteins where they serve as sensing domains or as mediators of protein–protein interactions [15]. CC domains are versatile structural motifs that vary greatly in both size and function, but are commonly found C-terminal to PAS domains where they are thought to contribute to signal transduction between the PAS domain and an effector domain [15,16,17]. The sGC CAT domain is a class III nucleotide cyclase domain and is the site of cGMP formation from guanosine triphosphate (GTP) [18]. A single active site is formed at the α/β dimer interface, with critical catalytic residues contributed by each subunit [19,20,21,22]. A second, catalytically inactive nucleotide binding site, termed the pseudosymmetric site, is also present at the CAT domain interface and is proposed to be involved in allosteric activity regulation by adenosine triphosphate (ATP) [23,24,25].

## 3. Activity Profile of sGC

In vitro, sGC exhibits a low basal level of activity in the absence of any ligand (Figure 1B) [23,26,27]. An increase in sGC activity is observed upon the addition of NO, either in the form of NO gas or through the use of small molecule NO donors, such as NONOates [23,26,27,28]. Interestingly, the extent of enzyme activation depends on the relative amount of NO added. A modest (~4-fold) increase in sGC activity is observed in the presence of a single molar equivalent of NO, which is termed the 1-NO state (Figure 1B) [23,26,27]. In this state, NO forms a 5-coordinate Fe^2+^–NO complex with the heme cofactor of the β H-NOX domain, which is discernable by ultraviolet-visible (UV-Vis) absorption spectroscopy as a shift in the Soret band of the heme from ~430 nm (for the Fe^2+^ unliganded species; the exact λ_max_ depends on the species of sGC) to 399 nm (Figure 1C) [29,30]. The activity of the 1-NO state accounts for, on average, ~15% of the maximal observed sGC activity, which is achieved in the presence of a stoichiometric excess of NO, and which is termed the excess-NO state (Figure 1B) [23,26,27]. In this state, the heme remains as a 5-coordinate Fe^2+^–NO complex and the site (or sites) of additional NO binding remains unknown, although cysteine residues have been implicated as potential binding partners [27]. Notably, the additional NO present in the excess-NO state can be removed by dialysis or buffer exchange over a desalting column to yield the 1-NO state of the protein, indicating that the additional NO forms only a transient interaction with sGC. After removal of the excess NO, the enzyme returns to the 1-NO state activity level.

In addition to the excess-NO state, a high level of sGC activity can also be achieved through the addition of small molecule stimulators to the 1-NO state of the protein. The prototypical sGC stimulator is the benzylindazol derivative YC-1 (3-(5′-hydroxymethyl-2′-furyl)-1-benzylindazole), which was first identified in a screen of compounds for platelet inhibition [31]. Since its discovery, the structure of YC-1 has been used as a template for drug development efforts aimed at creating sGC stimulators, including for the drug riociguat (Bayer’s Adempas^®^) that is used in the treatment of pulmonary hypertension [9,32,33,34,35]. Addition of YC-1 to the 1-NO state of sGC leads to activity that is comparable to that of the excess-NO state (Figure 1B) [27,36].

The existence of various activity states for sGC in vitro begs the question of physiological relevance—which activity states of sGC are relevant in vivo? In cells, the concentration of NO has been estimated to be in the range of 100–5000 pM [37], whereas estimates for the dissociation constant of NO from the sGC heme put the *K*_d_ in the low picomolar range, or lower [38]. Based on these considerations, the unliganded state of sGC is unlikely to exist in cells. Therefore, the physiologically relevant forms of sGC likely cycle between a 1-NO state and an excess-NO activated state. This inference is supported by two independent studies done in vein and aortic endothelial cells in which YC-1 was shown to stimulate cellular cGMP production to a high level, but not in the presence of inhibitors of nitric oxide synthase (NOS) [27,39]. The underlying assumption is that when NOS is inhibited, the intracellular NO concentration is low enough that sGC exists in the unliganded state. These results are consistent with the in vitro observations that YC-1 stimulates the 1-NO state of the protein to a high activity level, whereas the unliganded state is stimulated to a much lesser extent (Figure 1B).

## 4. Crystal Structures of Individual sGC Domains

Although more than four decades of work have revealed much about the biochemistry of sGC, obtaining a structure of the full-length enzyme proved elusive. In lieu of such information, structural insights on sGC came from crystal structures of individual sGC domains and their homologs, as summarized below.

### 4.1. The H-NOX Domain

Efforts to better understand the effect of NO binding to sGC turned to bacterial homologs of the β H-NOX domain. In aerobic and facultative anaerobic bacteria, NO-sensing H-NOXs are found as standalone proteins that interact with cognate signaling partners, usually a histidine kinase or a diguanylate cyclase [40,41]. In contrast, obligate anaerobes contain O_2_-sensitive H-NOX domains fused to C-terminal effector domains, generally a methyl-accepting chemotaxis protein [40,42]. Both NO-binding and O_2_-binding H-NOX domains have been studied in detail structurally and spectroscopically to provide insight into the function of this domain in sGC. The first H-NOX structure to be solved was of the O_2_-binding H-NOX domain from the thermophilic anaerobe *Caldanaerobacter subterraneous* (*Cs*) (previously known as *Thermoanaerobacter tengcongensis*) [43,44]. The structure revealed the H-NOX domain to consist of seven α-helices (labeled αA–G) and a four-stranded antiparallel β-sheet (with strands labeled β1–4). The *b*-type heme cofactor is found sandwiched between the N- and C-terminal subdomains of the protein and is ligated by a conserved histidine residue, termed the proximal histidine, which is located on the αF helix of the protein. The N- and C-terminal H-NOX subdomains are termed distal and proximal, respectively, with the proximal subdomain taking its name from the presence of the proximal histidine residue within the subdomain. The distal subdomain consists of α-helices A–E and the proximal subdomain is composed of helices αF and αG and the four-stranded β-sheet.

Subsequent crystal structures of the NO-binding H-NOX from *Shewanella oneidensis* (*So*) in the unliganded and NO-bound states revealed the structural effects of gas binding to the protein [45]. Upon NO binding, the protein undergoes a global conformational change in which the distal subdomain rotates ~4° relative to the proximal subdomain (Figure 2A). This rotation is centered on two conserved glycine residues within the αD and αG helices, called the glycine hinge [45,46,47]. A similar ligation state—induced conformational change was later observed in the O_2_-binding *Cs* H-NOX, and it has been suggested that this rotation of the distal subdomain may be a general feature of H-NOXs which could play a key role in signaling [46].

In addition to the overall conformational change, the structures also show NO-induced rupture of the iron–histidine bond between the heme cofactor and the protein, consistent with previous spectroscopic data [45]. This bond cleavage is accompanied by a slight distortion of the αF helix such that the Cα atom of the proximal histidine is displaced by ~2 Å and the side chain adopts an alternative rotamer conformation. Together, these movements result in an apparent ~90° rotation of the histidine side chain away from its position when it is ligated to the heme (Figure 2B). Notably, in the structure of the NO-bound *So* H-NOX, NO was observed bound to the proximal side of the heme, rather than to the distal side as had been expected. Subsequent spectroscopic work, however, revealed this phenomenon to be a result of the large excess of NO used in the crystallization conditions and that under standard solution conditions NO primarily binds to the distal side of the heme [48].

### 4.2. The PAS and CC Domains

PAS and CC domains are ubiquitous and versatile structural motifs that are found across domains of life and that often function together in a signaling capacity. PAS domains are named for the proteins Per, Arnt, and Sim, the three proteins in which homologous sequence regions, now recognized as the PAS domain, were first identified [49]. There are now over 40,000 annotated PAS domains in the Pfam database [50] and, despite having low sequence similarities on average, these domains share a common core fold consisting of a five-stranded antiparallel β-sheet flanked by several α-helices [15]. A subset of PAS domains bind cofactors and/or exogenous ligands and serves in a signaling capacity as direct sensors. For the majority of PAS domains, however, no ligand has been identified and these domains are thought to play a role in signal transduction and modulating protein–protein interactions [15]. Indeed, the sGC PAS domains are not known to bind a ligand but do contribute to dimerization. The first structural information on the sGC PAS domains came from a crystal structure of a histidine kinase PAS domain that is highly homologous to the sGC domains [51]. This structure showed that the PAS domains in sGC likely form a parallel dimer through a conserved dimerization motif that is common to many PAS domains [51,52]. Subsequently, a crystal structure of the α PAS domain from the *Manduca sexta* (moth) sGC was determined [53]. How the PAS domains may contribute to signal transduction in sGC is unclear from the structures, although rearrangement of internal hydrogen bonding networks has been implicated in structurally similar PAS dimers [52]. Hydrogen/deuterium exchange mass spectrometry (HDX-MS) revealed NO-induced differences in H/D exchange rates in parts of the PAS domains in the full-length *Rattus norvegicus* (rat) sGC, consistent with the idea that perturbations in this region occur upon activation [54].

As the name implies, CC domains are composed of two or more α-helices that intertwine, forming a coil of coils. These domains are found in a wide range of proteins and serve in a variety of functions. As in sGC, CC domains are often found C-terminal to PAS domains, where they are thought to serve as an intermediary signal transduction module between the PAS domain and a connected effector domain, this effector domain being the CAT domain in sGC [15]. Previous structural information on the sGC CC domain came from a crystal structure of the isolated rat β CC domain [55]. Efforts to obtain a structure of the α/β CC heterodimer in this study were stymied by an inability to express the α CC domain. The structure of the β CC showed a tetrameric assembly composed of two CC homodimers, and within each dimer, the two helices interact in an unexpected anti-parallel fashion. Based on sequence considerations and modeling of the full-length sGC structure, the authors reasoned that the anti-parallel arrangement likely does not reflect the true structure of the α/β CC heterodimer, which they concluded is more likely to be oriented in a parallel manner. This conclusion was supported by a later study in which chemical crosslinking between lysine residues in the moth sGC was more consistent with a parallel arrangement of the CC domains [56]. Similar to the PAS domains, the CC domains of the full-length rat sGC also exhibit changes in H/D exchange rates upon NO binding, consistent with alterations in this portion of the structure upon activation [54].

### 4.3. The CAT Domain

The first nucleotide cyclase CAT domains to be structurally characterized were those of adenylate cyclases (ACs) [57,58,59]. The AC CAT domain is comprised of a large eight-stranded mixed β-sheet surrounded by five α-helices (Figure 3A). Two CAT domain monomers intertwine in a head-to-tail fashion to form a wreath-like dimer (Figure 3B). The active site is found in a cleft at the dimer interface and is composed of residues that are contributed by each subunit. This architecture gives rise to two nucleotide/ligand binding sites at the dimer interface, which translates to the presence of two active sites in AC homodimers and of one active site and one pseudosymmetric site in AC heterodimers. In heterodimers, the pseudosymmetric site lacks the residues necessary for catalysis and is proposed to be involved in activity regulation [57,58].

For many years, AC CAT domain structures served as a basis for understanding the structure of GCs, given their high sequence similarities. Indeed, homology models of the bovine and rat sGC CAT domains that were made based on AC CAT domains were used to identify key residues involved in catalysis and substrate specificity [60,61]. Using these models as a guide, the authors in one study were able to alter the substrate specificity of the rat sGC, rendering it capable of performing AC activity [61]. More recently, crystal structures have been solved of both homo- and hetero-dimeric GC CAT domains, confirming that their structures are in fact similar to those of ACs [19,20,21,22].

To date, all GC CAT domain crystal structures represent inactive conformations in which the active sites are not appropriately oriented for substrate binding and subsequent catalysis [19,20,21,22]. Mechanisms for activation have been proposed based on structures of AC CAT domains that appear to represent catalytically competent conformations [59]. In general, these mechanisms entail a rotation of one subunit towards the other in a clamp like motion that involves movement of the β7–β8 loop and helices α1 and α2 [20,21,22]. In addition to this large-scale conformational change, smaller changes within the active site open up the substrate binding site. In particular, helix α4 shifts away from the central cleft to make room for substrate. Full closure of the active site is thought to occur after initial substrate binding in a step that brings key amino acid residues and the nucleotide substrate into favorable alignment for catalysis [19,20]. It has been suggested that substrate discrimination between GTP and ATP occurs during this closure step, as residues involved in base recognition are brought into position to optimally hydrogen bond with either guanine or adenine [19].

## 5. Low-Resolution Structures of Full-Length sGC

The first direct insight into the structure of full-length sGC came in 2014 with the report of low-resolution negative stain electron microscopy (EM) envelopes of the rat enzyme [62]. Three-dimensional EM maps of the uranyl formate-stained particles were obtained through random conical tilt (RCT) reconstructions and ranged in resolution from 25 to 40 Å. The maps revealed a dumbbell-shaped structure composed of two globular lobes connected by a thin, stalk-like region of density. The larger of the two globular lobes was assigned as the H-NOX/PAS bundle, with the two PAS domains interacting to form a dimer interface and the two H-NOX domains flanking either side of this central PAS dimer. The stalk-like structure extended from the PAS dimer and was assigned as the CC domains, running in a parallel orientation into the smaller globular lobe, which was assigned as the CAT domain dimer.

The single-particle classification and alignment in this study yielded a total of 94 3D reconstructions of the unliganded rat sGC that suggested significant conformational flexibility [62]. The reconstructions ranged from fully extended states, in which the H-NOX/PAS bundle and the CAT domains were separated by 40 Å, to more compact states, in which connective density was observed between the β H-NOX and CAT domains. The two ends of the CC dimer were described as pivot points around which these conformational changes take place. Notably, this conformational flexibility was independent of the enzyme ligation state, as structural analysis performed on NO and GTP analog-bound sGC revealed similar distributions of conformational states [62].

This low-resolution analysis of full-length sGC unveiled for the first time the overall shape and general domain organization of the enzyme, however, the mechanism of allosteric activity regulation remained unclear. It was not until the determination of higher resolution structures that these details began to become more apparent.

## 6. High-Resolution Cryo-EM Structures of Full-Length sGC

Within the last decade, the so-called “Resolution Revolution” [63] in single-particle cryo-EM has ushered in a new era in structural biology, and in 2019, two independent reports of high-resolution full-length sGC structures were published [64,65]. In the first report, Horst et al. presented structures of the sGC from *Manduca sexta* (moth) in the unactivated, Fe^2+^-unliganded state and in the excess-NO and YC-1-bound activated state to 5.1 and 5.8 Å resolution, respectively [64]. The second report, by Kang et al., presented several structures of the *Homo sapiens* (human) sGC in different states: the Fe^2+^-unliganded state (to 4.0 Å resolution), the Fe^3+^-unliganded state (3.9 Å resolution), the excess-NO activated state with bound Mg^2+^ and GMPCPP (3.8 Å resolution), and the β H105C variant of the enzyme (6.8 Å resolution) [65]. These two sGC isoforms share 38% sequence identity in the α subunits, 61% identity in the β subunits, and both exhibit the three-state activity profile discussed above [64,66,67]. Consistent with the similarities in sequence and activity, the structures of the moth and human enzymes are very alike and both sGCs were found to undergo a dramatic conformational change upon ligand binding and activation, as will be discussed in detail below. The Fe^3+^-unliganded form of human sGC was found to be structurally similar to the Fe^2+^-unliganded form and the β H105C variant, in which the heme-ligating histidine is replaced with cysteine, was found to lack the heme cofactor and to be similar in both structure and activity to the activated state of the enzyme [65].

### 6.1. General Architecture of Full-Length sGC in the Unactivated and Activated States

Consistent with the low-resolution, negative stain EM reconstructions, the high-resolution cryo-EM data reveal an oblong particle composed of two distinct lobes connected through a stalk-like density. The N-terminal regulatory lobe consists of the α and β H-NOX domains and the α/β PAS dimer (Figure 4). The two CC domains extend from the PAS dimer to the C-terminal regulatory lobe, which contains the α/β CAT domain dimer (Figure 4).

In the basal, unactivated state, sGC adopts a contracted conformation in which the β H-NOX domain is in close proximity to, although not directly interacting with, the β CAT domain (Figure 4A). This bent conformation appears to be achieved through an unexpected bending of the CC domains (highlighted in purple in Figure 4A). Following the PAS domain dimer, the α and β CC domains form helical segments of 3 and 2 turns, respectively, followed by loop regions that lead into two longer helical segments at an angle of ~90° from the initial short helices. Upon activation, the conformation of the protein extends as the regulatory lobe swings through a rotation of ~70° relative to the catalytic lobe (Figure 4B). In the activated state, the portions of the CC domains that are bent in the unactivated state have adopted helical folds to form two long, continuous helices from the PAS dimer through to the CAT dimer, resulting in the overall extended conformation (Figure 4B).

Interestingly, the structures do not indicate that there is any direct contact between the regulatory and catalytic lobes. Such direct contact had previously been suggested as the means of activity regulation in sGC, with the β H-NOX domain interacting with the CAT domains in an inhibitory fashion that was relieved upon NO binding [18,68]. Instead, it seems that the activation mechanism involves more subtle allosteric changes that are communicated through the length of the protein. The bending and straightening of the CC domains appears to be a key conformational change in this process, although the specific amino acid interactions that lead to this change remain unclear.

The high-resolution structures of full-length sGC in both the unactivated and activated states provide a first exciting look at the structural details of this important enzyme. They also set the stage for future experiments aimed at further understanding the mechanism of allosteric, intramolecular signal transduction. Aspects of individual domains that have been revealed by these new structures will be discussed in more detail below. The higher resolution of the human sGC structures allowed visualization of many amino acid side chains throughout the electron density maps, and so references to residue numbers will reflect the numbering of the human sGC sequence.

### 6.2. The α H-NOX Domain

The α H-NOX domain of sGC has long been an enigma. It has previously been shown to be unnecessary for dimerization or activity regulation, and yet is present in all known sGCs [14]. Despite sharing sequence similarity with the β H-NOX domain (Figure 5A), the α H-NOX does not bind heme, and has previously been termed a “pseudo–H-NOX” for this reason [14,54]. Prior to the high-resolution full-length sGC structures, no structural information on the α H-NOX domain was available, due at least in part to an inability to express the isolated domain in a soluble form [18]. Explanations for why this domain does not bind heme were based on sequence considerations—the heme ligating histidine residue is missing in the human sGC α sequence, whereas a conserved YxSxR motif that stabilizes the heme propionate chains in heme-binding H-NOXs is missing in the sequence of the moth sGC α subunit (Figure 5A). It was reasonably assumed that the lack of these residues must be the reason that the α H-NOX does not bind heme. With structures now in hand, however, the inability to bind heme becomes clear: an N-terminal extension (residues α-69–85, α-1–68 are disordered) of the α H-NOX runs into the heme pocket, occluding heme binding (Figure 5B). Although the function of this domain remains elusive, there is now a more complete understanding of why sGC contains only one heme per heterodimer.

### 6.3. The β H-NOX Domain

In contrast to the α H-NOX domain, the β H-NOX domain and its bacterial homologs have been well-characterized in previous studies. Much of the interest in the β H-NOX domain lies in the fact that it contains the binding site for the first equivalent of NO and is therefore an essential component of NO-dependent activity regulation. Previous structural work on bacterial H-NOX homologs revealed a ligation-state dependent conformational change in which the distal subdomain rotates relative to the proximal subdomain upon NO binding, as discussed above (Figure 2A) [45]. In the cryo-EM structures of full-length sGC, relative shifts between the two subdomains are less apparent (Figure 6A). When the β H-NOX domains of the unactivated and activated human sGC structures are aligned by the β sheet region of the proximal subdomain (equivalent to the alignment performed for the *So* structures in Figure 2A), helices αA–C of the distal subdomain are nearly superimposable (Figure 6A). Instead, a lateral shift is observed in the αF helix (residues β-94–112), which contains the heme ligating histidine residue (β-His105) (Figure 6A). This shift is likely to be induced by rupture of the iron–histidine bond, which occurs upon NO binding, and stands in contrast to the rotation of the αF helix that was observed in the structures of *So* H-NOX. Also in contrast to the *So* H-NOX structures is the relative positioning of β-His105 (His103 in *So* H-NOX) in the unactivated (unliganded) and activated (NO-bound) structures. Rather than the histidine side chain flipping away from the heme, the rotamer conformation of β-His105 is maintained following iron–histidine bond cleavage, while the Cα atom of the residue shifts by 2.1 Å (Figure 6A). Notably, if β-His105 were to flip upon NO binding as it does in *So* H-NOX, it would clash with β-Arg258 in the β PAS domain of the activated conformation of full-length sGC. Together, these differences in NO-induced structural changes suggest that the mechanism of communicating the heme ligation state is different for isolated H-NOX domains than it is for full-length sGC. Structures of bacterial H-NOX domains in complex with their cognate signaling partners would provide more insight into these differences.

One interesting aspect of the NO-induced shift of the αF helix is that it results in a loss of hydrogen bonding interactions between residues on the C-terminal end of the αF helix and residues within the α CC domain (Figure 6B). In particular, β-Thr110 appears to make contacts with α-Gln418 and α-Arg428 in the structure of the unactivated human sGC (Figure 6B). Additionally, β-His107 contacts α-Glu417 (Figure 6B). Notably, α-Gln418 and α-Arg428 bookend the bend in the α CC domain. Thus, one mechanism for NO-induced sGC activation may be that the shift of the αF helix disrupts contacts with these residues, allowing the α CC to straighten. The residues β-His107, β-Thr110, α-Glu417, α-Gln418, and α-Arg428 are all highly conserved in over 200 sGC sequences, underscoring their potential functional relevance. In previous biochemical work, a β-T110A variant of rat sGC was shown to exhibit higher basal activity than WT, consistent with a role for β-Thr110 in stabilizing the unactivated conformation of the protein [69]. A separate study, however, showed that the β-T110R variant had comparable activity to WT in enriched lysate assays, and so the exact function of this residue remains poorly understood [70]. Higher resolution structures and/or further mutagenesis studies would provide more insight into the role of β-Thr110 and the residues that it appears to interact with in the unactivated conformation.

Also on the αF helix is β-Asp106, which is also highly conserved in over 200 sGC sequences and has been implicated in sGC activation. Mutation of this residue to alanine or lysine has been shown to result in an sGC variant that cannot be fully activated by excess NO [65,70]. Interestingly, β-Asp106 appears to participate in a network of charge–charge interactions in both the unactivated and activated states, although the interaction partners are different in each case (Figure 6C,D). In the unactivated conformation, β-Asp106 contacts β-Arg258 in the β PAS domain, which in turn also contacts β-Asp374 in the β CC domain (Figure 6C). Activation of sGC and shifting of the αF helix disrupts these contacts, and in the activated state β-Asp106 contacts both β-Arg305 and β-Lys307 in the β PAS domain; β-Lys307 additionally interacts with α-Asp411 in the α CC (Figure 6D). Following disruption of its interaction with β-Asp106, β-Arg258 forms an interaction with β-Asp102 on the αF helix in the activated conformation (Figure 6D). Again, each of the residues involved in these interactions is highly conserved in sGCs. Together, the structural and mutagenesis data on β-Asp106 suggest that this residue plays an important role in stabilizing the activated conformation of sGC.

### 6.4. The PAS Domains

The α and β PAS domains of sGC form a parallel dimer, as predicted by the crystal structure of the homologous histidine kinase PAS domain [51]. The dimer interface is stabilized by a conserved dimerization motif consisting of hydrophobic interactions along an N-terminal amphipathic helix (residues α-279–286, β-210–217) as well as an adjacent β-strand (residues α-378–382, β-318–322) (Figure 7) [51,52]. The overall structure of the sGC PAS dimer changes very little upon activation (Figure 7). It was previously suggested that signal transduction in similar PAS domain dimers occurred through alterations of internal hydrogen bonding networks [52]. What these alterations are in sGC is unclear from the structures and how the PAS domains might play a role in signaling remains ambiguous. It may be that the PAS domains play a dynamic role in facilitating structural transitions that are not apparent in the static endpoint snapshots captured by EM. Alternatively, the role of the PAS domains could be mainly structural in facilitating dimerization.

### 6.5. The CC Domains

The most obvious conformational change that takes place upon sGC activation occurs in the CC domains. In the unactivated structure, each CC domain is divided into two nearly perpendicular helical segments connected by a loop region that straightens upon activation to form two long helices (Figure 8A). These bent loop regions comprise residues α-419–427 and β-355–359. The higher resolution (4.0 Å) structure of the unactivated human sGC allowed for more accurate modeling of the CC register in this region due to the ability to observe electron density for many of the side chains [65]. Consistent with the adoption of secondary structure, these residues were previously shown to exhibit decreased hydrogen–deuterium exchange upon sGC activation [54].

Since there is no direct contact between the regulatory H-NOX/PAS lobe and the catalytic lobe of sGC, structural changes in the CC domains are likely a key driver of allosteric signal transduction. Using their structural observations, Kang et al. engineered single proline mutations into the CC domains of human sGC (α-D423P or β-G356P) to generate a constitutive break in one or the other of the CC helices [65]. Both variants exhibited impaired NO-dependent activation, while the same residues mutated to alanine had no effect on activity [65]. Together, these experiments support the hypothesis that straightening of the CC domains is critical to sGC activation.

In addition to straightening, the CC helices also rotate about one another upon activation (Figure 8A). When the unactivated and activated structures are superimposed along the α CC domain, the β CC helix undergoes an apparent ~70° rotation around the α CC helix (Figure 8A). This rotation shifts the C-terminal ends of the CC helices (Figure 8B), which is likely important for transmitting signal to the CAT domains, as discussed below.

To stabilize the fully active form of moth sGC, Horst et al. included the small molecule sGC stimulator YC-1 in their EM grid preparation [64]. Electron density consistent with the size and shape of YC-1 was observed in the structure between the β H-NOX domain and the α CC domain. Two orientations of YC-1 were possible based on the observed density (Figure 8C). When the CC domains of moth sGC are modeled in the correct register, using the structure and sequence of human sGC as a guide, the YC-1 density sits along the stretch of the α CC that is bent in the unactivated enzyme (Figure 8C). Although the exact contacts between YC-1 and the protein are unclear given the limiting resolution of the structure (5.8 Å), the general positioning of YC-1 suggests that it, and related drug molecules like Adempas^®^ [9], may function by stabilizing the straightened conformation of the α CC domain.

### 6.6. The CAT Domains

As predicted from the crystal structures of isolated GC CAT domains, the CAT domains in full-length sGC form a wreath-like heterodimer that undergoes a conformational change upon activation [19,20,21,22]. This conformational change involves a clamp-like motion of the β subunit relative to the α subunit (Figure 9), as described above. Concomitant with the clamping of the β subunit, the active site opens through the movement of helix α4 of the β CAT domain (residues β-545–555) away from the α CAT domain (Figure 9). Kang et al. noted that the net result of these changes is an overall expansion of the central cleft between the two monomers from 1375 to 1549 Å^3^ [65]. Together, this opening of the central cleft and active site is consistent with the observed decrease in *K*_M_(GTP) upon sGC activation [71].

The conformational changes in the sGC CAT domains appear to be driven by upstream rearrangements within the protein. In particular, rotation of the CC helices brings the two C-terminal CC residues into closer proximity (Figure 8B), thus shifting the N-termini of the CAT domains. This shift then apparently propagates through the rest of the CAT domain architecture, ultimately resulting in the opening up of the substrate binding site.

## 7. Insight into the 1-NO State from Small Angle X-Ray Scattering

The cryo-EM structures presented by Horst et al. [64] and Kang et al. [65] represent conformations of sGC that correspond to the two extremes of enzymatic activity—the unactivated state and the fully activated state. sGC, however, exhibits a distinct three-state activity profile that includes an intermediary activity state, which is achieved through the binding of a single equivalent of NO to the sGC heme (Figure 1B). This 1-NO state is likely to be the physiological resting state, given the estimated intracellular NO concentration and the *K*_d_ of NO release from the heme, as discussed above [37,38]. To provide structural insight into this key state of sGC, Horst et al. employed size exclusion chromatography (SEC) coupled to small angle X-ray scattering (SAXS) to interrogate solution-state conformational changes [64]. The analysis revealed that the 1-NO state is best modeled as an ensemble, with 72% of the sample in the unactivated conformation and 28% of the sample in a partially extended conformation. The calculated structure of this partially extended conformation lies somewhere between that of the unactivated and fully activated structures observed by EM. This apparent conformational distribution explains the partial activity of the 1-NO state, which is, on average, ~15% of the maximum, fully activated sGC activity.

## 8. Conclusions

The structures reported by Horst et al. and Kang et al. represent a major advance in our understanding of sGC and provide a foundation for elucidating the biochemical details of sGC activation. In particular, the rearrangements that occur in the CC domains upon activation appear to be critical in communicating the ligation state of the protein to the CAT domains (Figure 4 and Figure 8). Other hints into the signal transduction process come from apparent contacts between conserved residues in the heme-containing β H-NOX domain and the α CC and β PAS domains (Figure 6). Together, these observations suggest a model for allosteric communication in sGC. Namely, shifting of the β H-NOX αF helix results in loss of contacts between the αF helix and the CC and PAS domains. These rearrangements promote straightening of the CC domains, which in turn induces shifts in the CAT domains that open up the GTP binding site. In this model, the activity states of sGC arise due to shifting of the equilibrium between bent and straight CC helices: basal activity is due to a small, fluctuating population of the unliganded enzyme that is in the elongated conformation. Binding of a first equivalent of NO to the heme cofactor induces higher activity through rupture of the iron–β-His105 bond, favoring the shift of the αF helix. The small molecule stimulator YC-1 shifts this equilibrium further towards high activity by stabilizing the straightened conformation of the α CC helix. Similarly, binding of additional NO to form the excess-NO state would also shift the equilibrium towards straightening of the CC domains, although the mechanism in this case remains unknown. Further mutagenesis studies are needed to provide biochemical support for this model and higher resolution structures of sGC bound to YC-1 would give insight into the contacts involved in stabilizing that interaction. Most importantly, identification of the site, or sites, of excess NO binding will be a critical step towards completing our understanding of sGC activation.

Additionally important is developing a better understanding of the 1-NO state of sGC, as this state is likely to be the physiological resting state of the enzyme. The SEC-SAXS analysis by Horst et al. suggested that the 1-NO state contains multiple conformations. Future work to characterize these conformations could include additional advanced SAXS techniques such as ligand titration SAXS coupled to singular value decomposition to identify structural transitions involved in forming the 1-NO state and in transition of the 1-NO state to the excess-NO state [72]. Additionally, performing cryo-EM on the 1-NO state may enable visualization of the different conformations present in the sample. In particular, advances in EM data processing techniques, such as the algorithm cryoDRGN, could facilitate the identification of discrete states and continuous conformational changes in heterogeneous samples [73]. Given these kinds of current, rapid advances in SAXS and EM analyses, there can be no doubt that understanding sGC from a structural perspective is only just beginning.

## Figures and Tables

**Figure 1 ijms-22-05439-f001:**
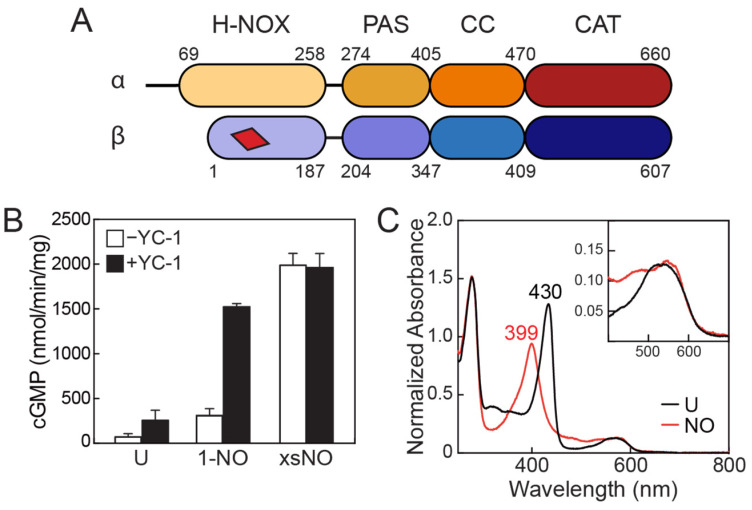
An introduction to sGC. (**A**) sGC domain architecture. sGC is a heterodimer composed of homologous α and β subunits. Each subunit contains four domains: an N-terminal heme NO/O_2_-binding (H-NOX) domain, a Per-Arnt-Sim (PAS) domain, a coiled coil (CC) domain, and a C-terminal catalytic (CAT) domain. The red trapezoid represents the heme cofactor found in the β H-NOX domain. Residue numbering of the domains corresponds to the sequence of human sGC. (**B**) Typical activity profile of sGC. sGC exhibits a low, basal level of activity in the absence of NO (U: unliganded). Addition of a single equivalent of NO (1-NO) results in a modest increase in activity. Full sGC activity is obtained in the presence of a molar excess of NO (xsNO). Activity can also be stimulated by the small molecule 3-(5′-hydroxymethyl-2′-furyl)-1-benzylindazole (YC-1). (**C**) UV-Visible absorption spectra of ligand binding to the sGC heme. Upon NO binding, the Soret peak shifts from ~430 to 399 nm. The inset shows the Q-band region of the spectra. Figure adapted from reference [11] [Creative Commons Attribution license].

**Figure 2 ijms-22-05439-f002:**
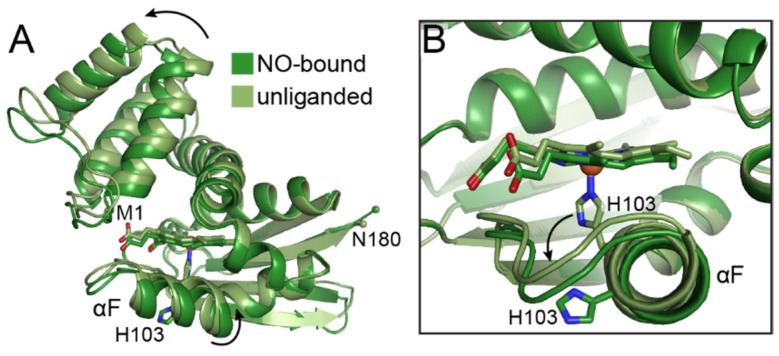
NO binding induces conformational changes in *So* H-NOX. (**A**) Overall structures of *So* H-NOX in the unliganded (light green, PDB ID 4U99 [45]) and NO-bound (dark green, PDB ID 4U9B [45]) states. NO binding induces rotation of the αF helix and a shift of the distal subdomain away from the heme binding pocket, as indicated by the black arrows. Structures were aligned by least squares fitting by the β-sheet region of the proximal subdomain (r.m.s.d. = 0.55 Å for 62 Cα atoms). (**B**) Zoomed in view of the heme cofactor showing cleavage of the iron–His103 bond as viewed down the axis of the αF helix. Protein is shown in ribbon representation with the heme in sticks and the iron center in spheres. His103 is shown in sticks. Sticks and spheres are colored with C in green, N in blue, O in red, and Fe in orange. The N- and C-termini are labeled and shown as small spheres in panel A.

**Figure 3 ijms-22-05439-f003:**
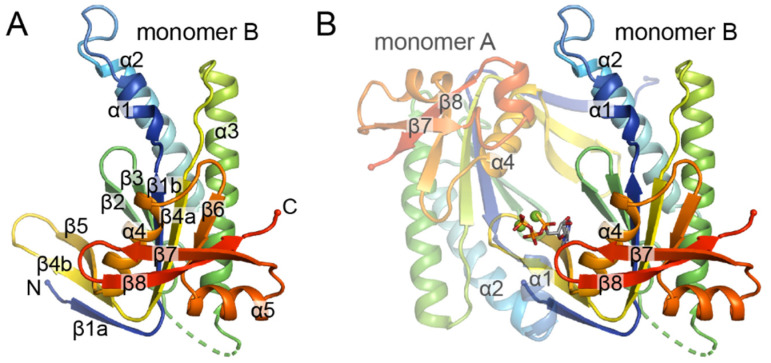
Crystal structures of AC CAT domains serve as a prototype for GC CAT domains. (**A**) Structure of the AC CAT domain monomer (PDB ID 1CJK [59]). Secondary structure elements are labeled according to conventions from the literature [57,58]. Protein is shown in ribbon representation and colored in rainbow mode from N- to C-terminus (termini are shown as small spheres and labeled) with the N-terminus in blue and the C-terminus in red. (**B**) Structure of the AC CAT dimer with ATP and two Mg^2+^ ions bound. Each monomer is shown as in panel A with monomer A at 50% transparency. Secondary structural elements that shift upon activation are labeled.

**Figure 4 ijms-22-05439-f004:**
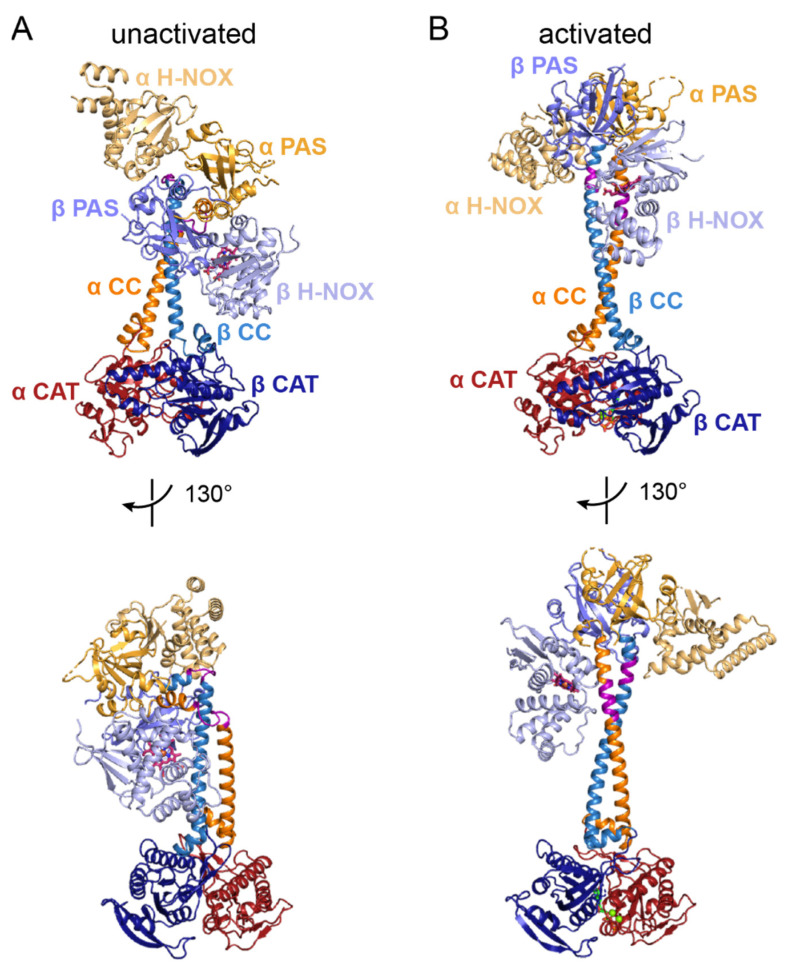
Activation of sGC induces a large-scale conformational change involving straightening of the CC domains. (**A**) Views of the unactivated conformation of sGC (PDB ID 6JT0 [65]). (**B**) Views of the activated conformation of sGC (PDB ID 6JT2 [65]). Protein is shown in ribbon representation with domains colored as indicated on the figure. The regions of the CC domains that bend and straighten are colored purple. The heme cofactor in the β H-NOX domain is shown in sticks with C in pink, N in blue, O in red, and Fe in orange.

**Figure 5 ijms-22-05439-f005:**
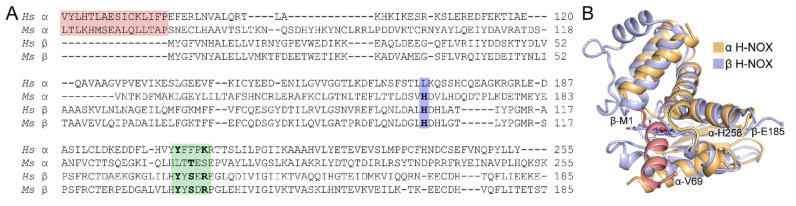
An N-terminal extension of the α H-NOX domain occludes heme binding. (**A**) Sequence alignment of the α and β H-NOX domains from human (*Hs*) and moth (*Ms*) sGCs. The α and β H-NOX domains share 20 and 18% sequence identity for the human and moth sequences, respectively. The N-terminal extension of the α H-NOX domain that is visible in the human sGC EM structure is highlighted in pink. The heme-ligating histidine (β-His105) and the YxSxR motif of the β H-NOX domain are highlighted in blue and green, respectively. Functionally conserved residues are in bold font. (**B**) Superposition of the α and β H-NOX domains from the unactivated human sGC structure (PDB ID 6JT0 [65]). The N-terminal helical extension of the α H-NOX domain is shown in pink and clashes with the heme in the β H-NOX domain. Structures were aligned by secondary structure matching (r.m.s.d. = 1.88 Å for 121 residues aligned). Protein is shown as in Figure 4 with the heme cofactor C atoms in light blue. The N- and C-terminal residues of each domain are labeled and shown as small spheres.

**Figure 6 ijms-22-05439-f006:**
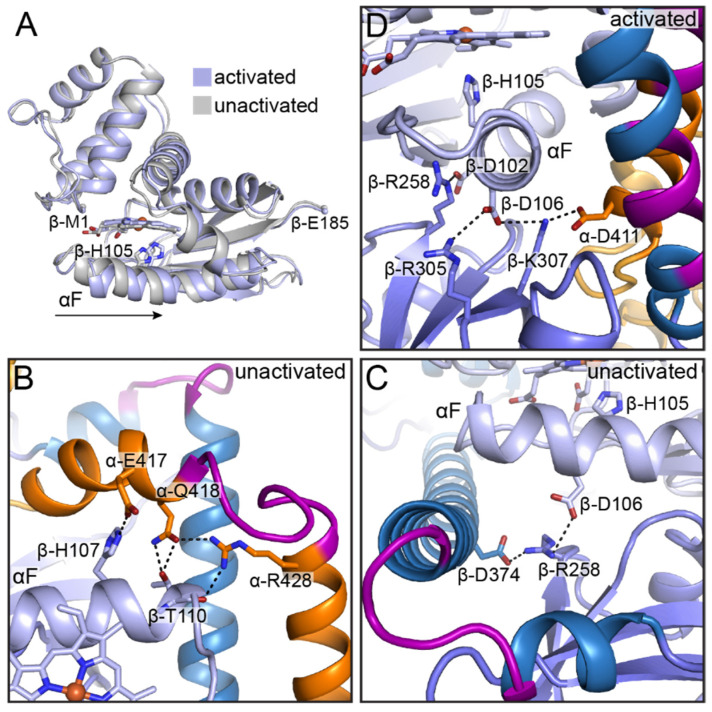
Structural rearrangements in the β H-NOX αF helix occur upon activation. (**A**) Superposition of the β H-NOX domains in the unactivated (grey, PDB ID 6JT0 [65]) and activated (light blue, PDB ID 6JT2 [65]) states. Activation induces a lateral shift of the αF helix, as indicated by the black arrow. Structures were aligned by least squares fitting by the β-sheet region of the proximal subdomain (r.m.s.d. = 0.70 Å for 67 Cα atoms). (**B**) In the unactivated state, residues within the αF helix make contact with residues in the α CC domain. (**C**) In the unactivated state, β-Asp106 on the αF helix participates in an interaction network with β-Arg258 in the β PAS domain and β-Asp374 in the β CC domain. The α sGC subunit has been omitted from this panel for clarity. (**D**) In the activated state, residues along the αF helix make different contacts than in the unactivated state. Domains are colored as in Figure 4 with the heme C atoms in light blue. Amino acid side chains are shown as sticks with C in respective domain colors, N in blue, and O in red. Hydrogen bonding interactions are shown as black dashes. The N- and C-terminal residues of the β H-NOX domain are labeled and shown as small spheres in panel A.

**Figure 7 ijms-22-05439-f007:**
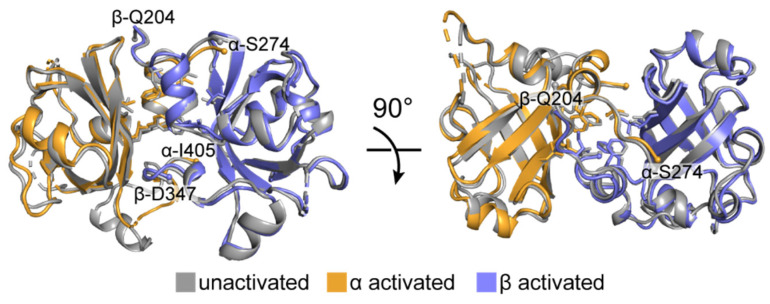
The two sGC PAS domains interact via a conserved dimerization motif and change very little upon activation. Superposition of the unactivated (grey, PDB ID 6JT0 [65]) and activated (colored as in Figure 4, PDB ID 6JT2 [65]) sGC PAS domains. The hydrophobic residues (α-I277, α-F282, α-F286, α-L380, α-L382; β-I208, β-F213, β-F217, β-L320, β-L322) that interact at the dimer interface are shown as sticks. Structures were aligned by secondary structure matching over both chains (r.m.s.d. = 0.96 Å for 234 residues aligned). The N- and C-terminal residues of each domain are labeled and shown as small spheres (C-terminal residue labels have been omitted from the bottom panel).

**Figure 8 ijms-22-05439-f008:**
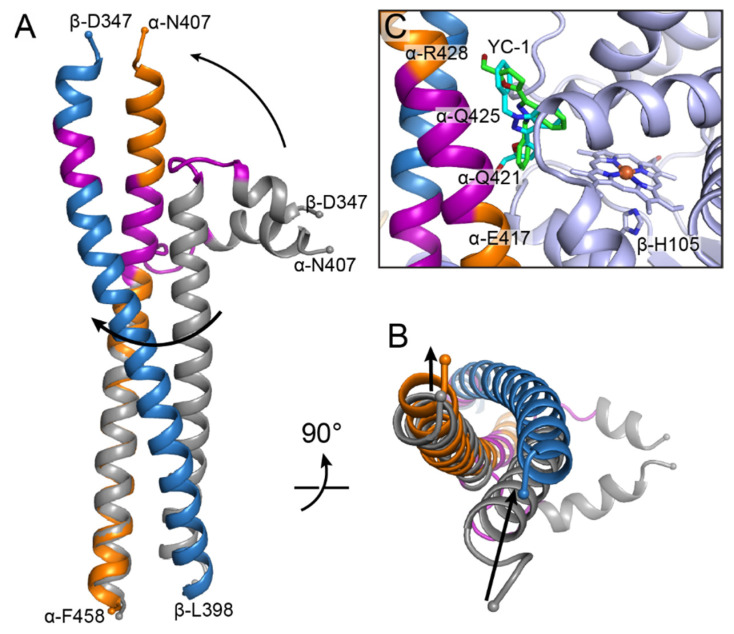
The CC domains play an important role in signal transduction within sGC. (**A**) Superposition of the CC domains in the unactivated (grey, PDB ID 6JT0 [65]) and activated (colored as in Figure 4, PDB ID 6JT2 [65]) states. The CC helices straighten upon activation concomitant with rotation of one helix relative to the other, as indicated by black arrows. Structures were aligned by least squares fitting of residues α429–α456 (r.m.s.d. = 1.11 Å for 28 Cα atoms). The N- and C-terminal residues of each domain are labeled and shown as small spheres. (**B**) View up the length of the CC domains from the C-terminal end. Rotation of the helices leads to a shifting of the C-terminal residues, as indicated by black arrows. (**C**) Electron density corresponding to YC-1 was observed between the β H-NOX and α CC domains. The two conformations of YC-1 that fit the density are shown in sticks with C in green and cyan, N in blue, and O in red. Residue numbers correspond to the human sGC sequence. β-His105 has been shown in sticks and labeled as a reference point for the αF helix.

**Figure 9 ijms-22-05439-f009:**
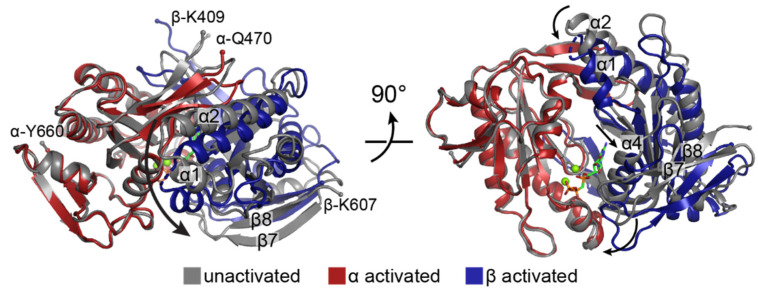
The sGC CAT domains undergo a conformational change upon activation. Superposition of the unactivated (grey, PDB ID 6JT0 [65]) and activated (colored as in Figure 4, PDB ID 6JT2 [65]) sGC CAT domains. Clamping of the β subunit relative to the α subunit is achieved through motions of the β7–β8 loop and the α1 and α2 helices, as indicated by the curved black arrows. Shifting of the α4 helix opens the active site for substrate binding, as indicated by the straight black arrow. The substrate analog GMPCPP of the activated structure is shown as sticks with C in green, N in blue, O in red, and P in orange. The two magnesium ions are shown as green spheres. Structures were aligned by least squares fitting of the α subunits (r.m.s.d. = 1.13 Å for 189 Cα atoms). The N- and C-terminal residues of each domain are labeled and shown as small spheres.

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
