# Peer review of "Structural Perspectives on the Mechanism of Soluble Guanylate Cyclase Activation"

_ijms, 2021, doi:10.3390/ijms22115439_

Round 1

Reviewer 1 Report

This review of Wittenborn and Marletta is a complete and interesting update on the structure of the guanylate cyclase enzyme involved in nitric oxide/cGMP messenger transduction pathway. The authors start with a description of the fundamental biochemical and structural characteristics of this family of enzymes; then they describe the first structural studies of the isolated domains that revealed the heterodimeric structure based on the analogy with the enzyme adenylate cyclase. Moreover, they describe the most recent evidences on the full-length structure obtained by  low-resolution negative stain electron microscopy and,  in 2019, by single-particle high-resolution Cryo-EM of whole enzyme describing the conformational changes of the enzyme in the inactive state, in the presence of low NO and fully activated. This high-resolution structure of full-length enzyme provide very important details of NO-responsive enzyme including the structural rearrangement of beta H-NOX domain, containing eme cofactor involved in enzyme activation. Finally, very interesting are also the data about the low NO concentration (1-NO state) using the SAXS technique.

The review is well written and a good fit for the journal. It is an interesting and comprehensive update on guanylate cyclase enzyme and may give useful insights to researcher working on enzyme structures.

Author Response

Thank you for your comments. We too hope the review will prove useful to those working on NO signaling and sGC structure.

Reviewer 2 Report

This is a full-length review article regarding the structure of soluble guanylyl cyclase and its mechanistic contribution to activation. The article includes early structural analyses by x-ray crystallography of clearly discernable structural domains of the enzyme and discussion of regulatory importance of the domain structures common in many proteins. The highlight of the article is the recently published structures of the full-size enzyme by cryo-EM. Thus, it is timely to publish this review article at this time. The structures of active and inactive state of enzyme are well described. Plausible activation mechanisms based on the structural changes and the roles of conserved amino acid residues are presented, many of which are only revealed by the cryo-EM study of the full-length protein. The article also points out the limitations of the current structural studies and future studies to further elucidate the activation mechanism of the protein. The article is very well written and good reading material for outsiders as well as new comers for the field.

The only minor error this reviewer noticed is font size mixed up in the legend for Figure 7 (L457-L464).

Author Response

Thank you for your comments. Our revised ms has fixed the font size issue you mentioned.